



# SICOPOLIS-AD v1: an open-source adjoint modeling framework for ice sheet simulation enabled by the algorithmic differentiation tool OpenAD

Liz C. Logan[1], Sri Hari Krishna Narayanan[2], Ralf Greve[3], and Patrick Heimbach[1,4,5]

[1]Oden Institute for Computational Science and Engineering, University of Texas at Austin, 201 East 24th Street, Austin, Texas, 78712, USA
[2]Mathematics and Computer Science Division, Argonne National Laboratory, Lemont, IL 60439, USA
[3]Institute of Low Temperature Science, Hokkaido University, Kita-19, Nishi-8, Kita-ku, Sapporo 060-0819, Japan
[4]Jackson School of Geosciences, University of Texas at Austin, 201 East 24th Street, Austin, Texas, 78712, USA
[5]Institute for Geophysics, University of Texas at Austin, J.J. Pickle Research Campus, Bldg. 196, 10100 Burnet Road (R2200), Room 2.236, Austin, TX 78758, USA

**Correspondence:** Patrick Heimbach (heimbach@utexas.edu)

**Abstract.** We present a new capability of the ice sheet model SICOPOLIS that enables flexible adjoint code generation via source transformation using the open-source algorithmic differentiation (AD) tool OpenAD. The adjoint code enables efficient calculation of sensitivities of a scalar-valued objective function or quantity of interest (QoI) to a range of important, often spatially varying model input variables, including initial and boundary conditions, as well as model parameters. Compared to

earlier work on adjoint code generation of SICOPOLIS, our work is based on several important advances: (i) it is embedded within the up-to-date trunk of the SICOPOLIS repository – accounting for one and a half decades of code development and improvements – and is readily available to the wider community; (ii) the AD tool used, OpenAD, is an open-source tool; (iii) the adjoint code developed is applicable to both Greenland and Antarctica, including grounded ice as well as floating ice shelves, and with an extended choice of thermodynamical representations. A number of code refactorization steps were required. They

are discussed in detail in an Appendix as they hold lessons for application of AD to legacy codes at large. As an example application, we examine the sensitivity of the total Antarctic Ice Sheet volume to changes in initial ice thickness, summer precipitation, and basal and surface temperatures across the ice sheet. Simulations of Antarctica with floating ice shelves show that over 100 years of simulation the sensitivity of total ice sheet volume to the initial ice thickness and precipitation is almost uniformly positive, while the sensitivities to surface and basal temperature are almost uniformly negative. Sensitivity to

summer precipitation is largest on floating ice shelves from Queen Maud to Queen Mary Land. The largest sensitivity to initial ice thickness is at outlet glaciers around Antarctica. Comparison between total ice sheet volume sensitivities to surface and basal temperature shows that surface temperature sensitivities are higher broadly across the floating ice shelves, while basal temperature sensitivities are highest at the grounding lines of floating ice shelves and outlet glaciers. A uniformly perturbed region of East Antarctica reveals that, among the four control variables tested here, total ice sheet volume is most sensitive to

variations in summer precipitation as formulated in SICOPOLIS. Comparison between adjoint- and finite-difference-derived





sensitivities shows good agreement, lending confidence that the AD tool is producing correct adjoint code. The new modeling infrastructure is freely available at www.sicopolis.net under the development trunk.

*Copyright statement.* The submitted manuscript has been created by UChicago Argonne, LLC, Operator of Argonne National Laboratory ('Argonne'). Argonne, a U.S. Department of Energy Office of Science laboratory, is operated under Contract No. DE-AC02-06CH11357.

The U.S. Government retains for itself, and others acting on its behalf, a paid-up nonexclusive, irrevocable worldwide license in said article to reproduce, prepare derivative works, distribute copies to the public, and perform publicly and display publicly, by or on behalf of the Government. The Department of Energy will provide public access to these results of federally sponsored research in accordance with the DOE Public Access Plan. http://energy.gov/downloads/doe-public-access-plan.

# 1 Introduction

Our ability to react to and mitigate the consequences of global sea level rise depends on the skill with which we may project change in the Earth's climate system. Central to characterizing and quantifying our uncertainty in expected outcomes is our understanding of ice sheet dynamics, their variability, their response to various climate forcing scenarios, and their simulated sensitivity to uncertain, empirical model parameters. A key component in the push to reduce this uncertainty has been the development of more sophisticated ice sheet models. Scientists have made significant strides, with the latest class of ice sheet

models resolving all three dimensions of the ice sheet's internal stress balance (as opposed to previous classes of models, which employed various approximations to the stress field to save computational cost). However, while advances in computational glaciology have enabled us to simulate ice sheet behavior more accurately, remaining uncertainties in the range of independent input variables required for ice sheet simulations, in particular initial conditions, surface forcings, basal boundary conditions, and internal parameters, comprise crucial weaknesses in ice sheet – and, ultimately climate system – prediction or projection

(Goelzer et al., 2018; Seroussi et al., 2019). As such, ice sheet modeling is facing similar issues of robust model initialization for prediction as those faced by the climate modeling community at large (e.g., Meehl et al., 2014; Balmaseda, 2017).

Ice flow critically depends on quantities that we either cannot easily measure (such as the friction or thermal forcing between ice and the bedrock below it), that parameterize subgrid-scale processes or empirical constitutive laws, or that we may never be able to measure in present day (such as the rate of snowfall in the past). These unknown or uncertain variables can be

construed as sets of parameters that we must infer if we are to make projections with ice sheet models, and these parameters must both satisfy, by some measure, the assumed model physics and the sparsely-made observations across such large bodies. In the language of optimal estimation and control theory, these parameters are referred to as control variables (The Analytic Sciences Corporation and Gelb, 1974).

If we wish to integrate ice sheet model projections into societally relevant discussions on sea level rise, we may wish to know

the sensitivity of key quantities of interest that represent integrated quantities of an ice sheet to the range of uncertain model inputs. For example, we wish to know how the total ice volume (above flotation) of an ice sheet is influenced by climatically relevant quantities (such as surface atmospheric forcings) or environmental variables (such as the melting on the bottom of





an outlet glacier or floating ice shelf that drains an ice sheet). A computationally costly method for deriving such sensitivities might use individually-made perturbations to the bottom melting rate at each location of the ice sheet's base, in particular at its margins that are in contact with ocean water. This means that the ice sheet dynamics must be integrated throughout time for each simulation experiment in which a point-wise perturbation has been applied in order to assemble a sensitivity map across

the entire domain to this control variable (basal melting). While the target of this approach remains of paramount importance – relating the output of an ice sheet model to poorly known inputs – the means are computationally expensive: understanding, for instance, the Antarctic Ice Sheet's sensitivity to changes in melting or basal friction means simulating the entire ice sheet throughout time for every perturbation made at each point in the domain. In this case the computational cost of such a method scales with the dimension of the domain grid, and as such is prohibitive.

Fortunately, adjoint models provide us with a means to this end where the computational cost of deriving sensitivity maps does not depend on the dimension of the control variable space. The adjoint model is in effect the transpose of the linearized operator of the ice sheet model. It propagates the dual of the ice sheet model state backward in time to simultaneously calculate the sensitivity of some quantity of interest (e.g., the volume of an ice sheet) with respect to some set of control variables (e.g., the basal melting beneath the ice, surface accumulation, or initial conditions). Thus, unlike the tangent linear model, which

computes the impact of *one input* perturbation on *all model outputs* at the cost of one execution (directional derivative), the adjoint model delivers the sensitivity of *one output* quantity of interest (QoI) with respect to *all model inputs*. This is useful not only for understanding the sensitivity of some scientifically or societally interesting quantity to model inputs, but further (and perhaps more interesting) enables the recovery of other forcing or initial conditions (e.g., initial ice sheet geometry or rate of snow accumulation throughout time and space) through formal inversion.

**1.1    Algorithmic Differentiation (AD) and its uses**

Generally, adjoint models arise in at least two classes of geophysical investigations:

**PDE-constrained, gradient-based optimization:**

Adjoint-enabled optimization problems may be posed in the following manner, beginning by formulating a scalar-valued cost function based on a least-squares model-data misfit and subject to prior information on the uncertainty of the control variables:

$$\mathcal{J} = [x_0 - x_{\mathrm{b}}]^{\mathrm{T}} C_{\mathrm{pr}}^{-1} [x_0 - x_{\mathrm{b}}] + \sum_{i=0}^{N} [y(t_i) - E_i(x(t_i))]^{T} C_{\mathrm{err}}^{-1} [y_i - E_i(x(t_i))], \tag{1}$$

where $x_0$, $x_{\mathrm{b}}$ and $x(t_i)$ are the initial, background and time-varying model state at time $t_i$, respectively. $y_i$ is the set of observations at time $t_i$ and $E_i(x(t_i))$ is a projection of the model state at time $t_i$ to space of observations $y_i$ (or the data-model misfit). $C_{\mathrm{pr}}$ is a prior error covariance matrix (usually diagonal as its structure is not fully known), and $C_{\mathrm{err}}$ is an observational

error covariance matrix (see Wunsch and Heimbach (2007) for a more comprehensive treatment).

PDE-constrained optimization seeks to find the gradient of $\mathcal{J}$ with respect to the control variables (here $x_0$), subject to the requirement that the (in general nonlinear) model $\mathcal{L}$ is fulfilled, rendered by a set of partial differential equations that step the





state $x$ from time $t_i$ to $t_{i+1}$, i.e.,

$$x(t_{i+1}) = \mathcal{L}(x(t_i)) \tag{2}$$

This problem is efficiently solved by means of the Lagrange multiplier method (Wunsch and Heimbach, 2007). The Lagrange multipliers have a direct interpretation as adjoint sensitivities or gradients of the cost function, equation (1), with respect to the
control variables, $\frac{\partial \mathcal{J}}{\partial x_0}$; they are used to seek a state of the system, $x^*$, that is tolerably close to the minimum of $\mathcal{J}$ As such, reliable adjoint values are essential in recovering the optimal $x^*$, which minimizes model-data misfit as presented by equation (1). This procedure is done in an iterative fashion, with an initial guess of the state $x_{\mathrm{b}}$ that is successively updated to achieve optimal boundary and initial conditions throughout the model's simulation. These optimal boundary and initial conditions are a part of the state space of $x^*$, in addition to more straightforward controls that describe the optimal state of the model (e.g., its
velocity, pressure, temperature, etc.).

    A model that can reproduce the optimal behavior of, e.g., an ice sheet throughout time possesses the advantage that model-derived predictions might be made with greater confidence, having been initialized by dynamics that are informed by spatio-temporal observations. In other words, the commonly-termed 'spin-up' of an ice sheet may produce more confident projections when forced by optimally recovered initial and boundary conditions, and an optimal state estimate, which may be recovered by
a time-dependent adjoint model. A model initialized and projected under such circumstances might better reproduce what can be inferred about its past state by observations, subject to the additional constraint of the assumed and (perhaps more subtle, but equally important) conserved model physics throughout time. Thus the constrained optimization problem of recovering boundary and initial conditions, and the model's optimal internal state dynamics throughout space and time, might be approached firstly by the task of obtaining reliable adjoint sensitivities; further, adjoint sensitivity analysis alone can be an elucidating
and worthwhile pursuit (the subject of the work presented here), by helping reveal how QoIs or costs, $\mathcal{J}$, are dependent on uncertainly known parameters in non-linear models, $F$.

**Sensitivity analysis:**

Beyond applications in optimization, the adjoint may also be widely applied to comprehensive analysis of linear sensitivities. Errico and Vukicevic (1992) and Marotzke et al. (1999) provide example applications in the context of atmosphere and ocean
modeling, respectively. For a general, scalar-valued function $\mathcal{J}$, now termed quantity of interest (QoI), the tangent linear model (TLM) $L$ of a given (in general nonlinear) model $\mathcal{L}$. eqn. (2) acts as directional derivative, as it propagates small perturbations in the control variable, $\delta x$ to corresponding perturbations in the QoI, $\delta \mathcal{J}$. In turn, the adjoint model (ADM), formally the transpose $L^T$ of the tangent linear model, propagates the sensitivities of the QoI to each component the control space, i.e. the partial derivatives of the augmented space of all control variables backward in time. Thus, whereas the TLM produces
directional derivatives of $\mathcal{J}$, the adjoint produces the gradient of $\mathcal{J}$. A detailed treatment in the context of ice sheet modeling is provided by Heimbach and Bugnion (2009) and Goldberg and Heimbach (2013). For the present purpose we summarize the way by which these sensitivities are formally obtained by way of algorithmic differentiation in the following.



## 1.2 Formal reverse mode of AD

The concept of the adjoint of a numerical model may be best understood in terms of the forward, original code construction and execution. If one wishes to know the sensitivity of some QoI (e.g., the volume of the Antarctic Ice Sheet) with respect to some model control variable (e.g., the average surface temperature in July), one method of pursuing knowledge about such

a sensitivity might be by perturbing the control variable, in sequence, at each single point within the discretized domain and propagating the perturbation forward in time. The perturbation to the control variable results in a change in the QoI, and one can proceed to calculate the sensitivity of the QoI with respect to the control variable everywhere in the domain. Herein these are termed the finite-differences of the QoI (or cost function) $\mathcal{J}$, with respect to the control variable $x$: $\frac{\Delta \mathcal{J}}{\Delta x}$, where $\mathcal{J}$ is calculated as in equation (1). An adjoint model code may be demonstrated as acceptable or reliable if the finite-difference-

derived sensitivities approximate the adjoint-derived sensitivities (within some tolerance); that is:

$$\frac{\Delta \mathcal{J}}{\Delta x_\varepsilon} = \frac{\mathcal{J}(x+\varepsilon) - \mathcal{J}(x-\varepsilon)}{2\varepsilon} \approx \frac{\partial \mathcal{J}}{\partial x} \qquad \forall \varepsilon \ll \| x \| \tag{3}$$

where $\varepsilon$ is the magnitude of the perturbation to the control and in this work varies depending on the examined control. Other finite-difference schemes may of course be employed, but for the purposes of this work we have selected the central difference for simplicity.

Adjoint models have been common in oceanic and atmospheric contexts (Talagrand and Courtier, 1987; Thacker and Long, 1988; Errico and Vukicevic, 1992) for decades. The method's popularity has been increasing steadily. MacAyeal et al. (1991) provides us with the earliest use of adjoint model-derived sensitivities of a simplified ice stream model. Observed velocities were used to invert for optimal basal friction parameters. While that study employed a simplified version of the Stokes equations and lacked time-dependence, other researchers have since undertaken the task of using adjoint sensitivities in a variety of ice-

related applications with more complexity (Vieli and Payne, 2003; Larour et al., 2005; Khazendar et al., 2007; Waddington et al., 2007; Joughin et al., 2009; Pattyn et al., 2008; Heimbach and Bugnion, 2009; Morlighem et al., 2010; Brinkerhoff et al., 2011; Goldberg and Sergienko, 2011; Gillet-Chaulet et al., 2012; Petra et al., 2012; Brinkerhoff and Johnson, 2013; Gagliardini et al., 2013; Goldberg and Heimbach, 2013; Morlighem et al., 2013; Larour et al., 2014; Perego et al., 2014; Isaac et al., 2015; Goldberg et al., 2015, 2016; Mosbeux et al., 2016).

An inherent problem in the numerical simulation of ice dynamics is the nonlinearity of the forward model. This arises due to the nonlinear dependence of viscosity on a stress or strain-rate formulation in ice. Because of this complication, hand-coded adjoints can be as labour-intensive (and error-prone) to develop as are their nonlinear parent model. As an alternative to hand-coding the adjoint model, algorithmic differentiation (AD) provides a method for obtaining adjoint codes via rigorous exploitation of the chain (and product) rule (Griewank and Walther, 2008; Forth et al., 2012; Naumann, 2012) (www.autodiff.org).

AD has been used in an array of applications in the geosciences and computational fluid dynamics, and has one substantive advantage to hand-written adjoint codes: it is flexible. Changes in the QoI defined, the control variables, or the underlying assumed and discretized model physics may lead to adjoint models of different structure. As models become more complicated due to time dependence and the inclusion of improved representation of ice physics, accurate, hand-coded adjoint solutions





may be ever more difficult to derive. In such contexts, AD methods provide a powerful alternative means for producing adjoint solutions to time-dependent problems that are up-to-date with respect to their parent forward model code.

Adjoint models developed by AD exploit the chain and product rules for the computation of derivatives of a function ($\mathcal{J}$) with respect to a set of input variables ($x$). To demonstrate how a scalar QoI, $\mathcal{J}$, is related to a control vector $x$, consider the following time-dependent statement of the problem, where $t \in (t_0, t_{\mathrm{f}})$ represents marching the model forward through discretized time steps:

$$u(t_{\mathrm{f}}) = \mathcal{M}(x) = \mathcal{L}_{N_t - 1}(\cdots (\mathcal{L}_1(\mathcal{L}_0(x)))) \tag{4}$$

where $u(t_{\mathrm{f}})$ is the model's state at the end of the simulation, $\mathcal{M}$ represents a mapping of control vector to the final state of the model, and $\mathcal{L}$ is the nonlinear system of equations (or forward model), applied successively to the initial state of the model. The subscripting of $\mathcal{L}$ refers to the time marching of the model, where $t_{\mathrm{f}} = \Delta t \, N_t$, and $\mathcal{L}_n$ maps the model state at time $n$ to $n + 1$. As our interest here is to show how gradients are generated by this method, consider then how linear perturbations to the control space result in changes to the cost function, $\mathcal{J}$ through a Taylor series expansion

$$\mathcal{J}(x_0 + \delta x) = \mathcal{J}(x_0) + \delta \mathcal{J} + \mathcal{O}(\delta \mathcal{J}) \tag{5}$$

Assuming $\mathcal{O}(\delta \mathcal{J})$ is negligible, $\delta \mathcal{J}$ is shown (in the forward sense) to be

$$
\begin{aligned}
\delta \mathcal{J} &= \langle \frac{\partial \mathcal{J}}{\partial x}, \delta x \rangle \\
&= \langle \frac{\partial \mathcal{J}}{\partial u(t_1)}, \frac{\partial u(t_1)}{\partial x} \delta x \rangle \\
&= \langle \frac{\partial \mathcal{J}}{\partial u(t_2)}, L_1 \frac{\partial u(t_1)}{\partial x} \delta x \rangle \\
&= \langle \frac{\partial \mathcal{J}}{\partial u(t_3)}, L_2 L_1 \frac{\partial u(t_1)}{\partial x} \delta x \rangle \\
&= \vdots \\
&= \langle \frac{\partial \mathcal{J}}{\partial u(t_{\mathrm{f}})}, L_{N_t - 1} \cdots L_2 L_1 \frac{\partial u(t_1)}{\partial x} \delta x \rangle
\end{aligned}
\tag{6}
$$

where $\langle \cdot, \cdot \rangle$ is the inner product and $L_n = \frac{\partial u(t_{n+1})}{\partial u(t_n)}$ is the *tangent linear model* of $\mathcal{M}$, a linearization of $\mathcal{L}$ at time $t_n$ about $x_0$. It follows from equation (6) that the *adjoint model* $L^{\mathrm{T}}$ (where $L^{\mathrm{T}}$ is the transpose of $L$) equivalently defines $\delta \mathcal{J}$:

$$
\begin{aligned}
\delta \mathcal{J} &= \langle \frac{\partial u(t_1)^{\mathrm{T}}}{\partial x} L_1^{\mathrm{T}} L_2^{\mathrm{T}} \cdots L_{N_t - 1}^{\mathrm{T}} \frac{\partial \mathcal{J}}{\partial u(t_{\mathrm{f}})}, \delta x \rangle \\
&= \langle \frac{\partial \mathcal{J}}{\partial x}, \delta x \rangle.
\end{aligned}
\tag{7}
$$

Equation (7) demonstrates that the sought gradient, $\frac{\partial \mathcal{J}}{\partial x}$, is computed by projecting the cost function to the model's final state, $\frac{\partial \mathcal{J}}{\partial u(t_{\mathrm{f}})}$, and mapping it *backward in time* ultimately to the dependence of the model on its (user-selected) control variables. Figure 1 presents a small example of the computational flow of the tangent linear (forward) and adjoint (reverse) modes of OpenAD applied to a single model, $\mathcal{F} : y = \sin(a * b) * c$, where the gradient of $\nabla \mathcal{F} = [\mathrm{a_d}, \mathrm{b_d}, \mathrm{c_d}]$ is sought.



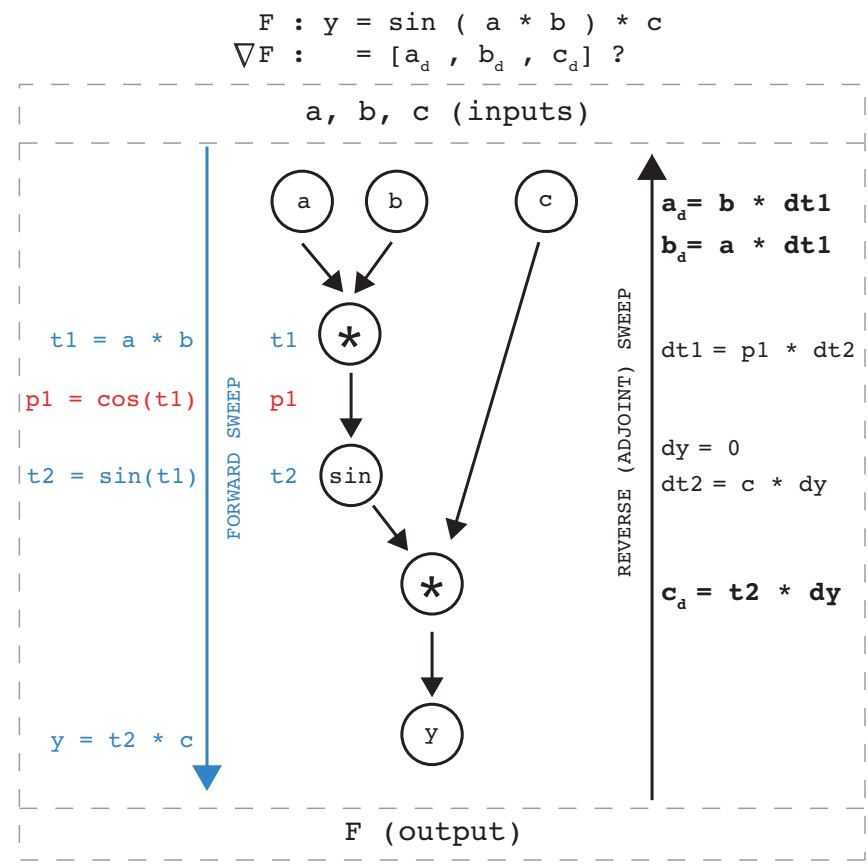

**Figure 1.** Schematic of AD applied to a simple function, $\mathcal{F}$. In SICOPOLIS-AD, the entire forward code is composed of many lines of simple functions like $\mathcal{F}$, in sequence. OpenAD provides $\nabla \mathcal{F} = [a_d, b_d, c_d]$ by relating the partials of `t1` and `t2` to intrinsically differentiable functions, like `sin()` (here the red text, `p1`). The partial derivatives of $\mathcal{F}$ are computed via the writing to memory of intermediate partial quantities, like `dy`, `dt1`, and `dt2`. Thus, the sought sensitivity of a QoI, $\mathcal{F}$, is related to input parameters, `a`, `b`, and `c` in this algorithmic (albeit much simplified) manner.

Any forward numerical model can be conceived of as a sequence or composition of simple operations like those shown in figure 1, with a single line representing one single algorithmic step. Via AD methods, then, the tangent linear and adjoint of a numerical model is provided by exhaustive application of the chain and product rules, line-by-line, to the model. The forward (section 2.1) or reverse (adjoint) mode of the model may be thought of as the composition in forward or reverse order of the

5  Jacobian matrices and their transpose of the full forward code's line-by-line algorithmic elements, that is, $L$ in eq. (6) and $L^{\mathrm{T}}$ in eq. (7).





## 2 Model description

### 2.1 Forward model SICOPOLIS

We begin with the ice sheet model SICOPOLIS (SImulation COde for POLythermal Ice Sheets) and develop its adjoint model from version 5-dev (Greve, 2019; Rückamp et al., 2019) (www.sicopolis.net). SICOPOLIS is open source and written in
Fortran; it has a relatively long and stable history (Greve, 1997). SICOPOLIS has remained a relevant and powerful tool for the cryosphere community and continues to participate in model intercomparison exercises (e.g., Goelzer et al., 2018; Seroussi et al., 2019). The model couples ice sheet dynamics and thermodynamics (solving for the ice thickness, extent, velocity, temperature, water content, and age) over three-dimensional domains including, among others, Greenland, Antarctica, and the polar caps of Mars. It employs simplified versions of the three-dimensional Stokes equation (internal stress balance), including:
the shallow-ice approximation for ice resting on land (Hutter, 1983; Morland, 1984); the shallow-shelf approximation for ice floating in the ocean (Morland, 1987; MacAyeal, 1989; Weis et al., 1999), and the shelfy-stream approximation for fast-flowing ice streams with limited coupling to the bed (Bernales et al., 2017). For a detailed treatment of the numerical methods employed in the model, readers are referred to Greve and Calov (2002); Greve and Blatter (2009, 2016); Bernales et al. (2017).

SICOPOLIS employs four different thermodynamics representations: (1) a two-layer, polythermal scheme, which allows for
the computation and effects of liquid water within a warmer, temperate layer; (2) a purely cold-ice scheme (in which no liquid water is present); (3,4) two flavours of the one-layer enthalpy scheme which combine the physical adequateness of (1) with the greater numerical simplicity of (2) (Aschwanden et al., 2012; Greve and Blatter, 2016). In all cases, horizontal diffusion of the thermodynamic fields (temperature, water content or enthalpy) is neglected. The solvers employed use an implicit discretization scheme for the vertical derivatives and an explicit scheme for the horizontal derivatives.

SICOPOLIS simulates ice as a nonlinear viscous fluid, employing Glen's flow law (Glen, 1955) amended as in Greve and Blatter (2009):

$$\eta = \frac{1}{2A(T')[\sigma_e^{n-1} + \sigma_0^{n-1}]},$$
(8)

where $\eta$ is the ice viscosity, $T'$ is the temperature difference relative to the pressure-melting point, $\sigma_e$ is the effective shear stress, and $\sigma_0$ is a small constant use to prevent singularities when $\sigma_e$ is very small. $n$ is the flow law exponent (taken as 3),
and $A$ is a temperature- and pressure-dependent rate factor (Cuffey and Paterson, 2010) that is modified in temperate regions containing liquid water following Lliboutry and Duval (1985).

Basal sliding under grounded ice links the sliding velocity, $v_b$, to the basal shear traction, $\tau_b$, and the basal normal stress, $N_b$ (counted positive for compression), in the form of a Weertman-type sliding law (e.g., Weertman, 1964; Budd et al., 1984):

$$v_b = -C_b \frac{\tau_b^p}{N_b^q},$$
(9)

where $C_b$ is the sliding coefficient, and $p$ and $q$ are the sliding law exponents.





## 2.2 Adjoint model of SICOPOLIS: algorithmic differentiation and OpenAD

As described in section 1.1, the construction of an adjoint model of a nonlinear, time-dependent forward model often presents a formidable task when solved analytically or hand-coded (e.g., Goldberg and Sergienko, 2011; Gillet-Chaulet et al., 2012; Morlighem et al., 2013; Isaac et al., 2015). In those works, the variational forward and adjoint equations are derived first and then discretized. As an alternative, AD produces adjoint code through differentiation of source code, using source-transformation tools. As the standard of numerical models (in various contexts) has risen to more complicated physical representations, the use of AD has become increasingly popular (e.g., Heimbach and Bugnion, 2009; Larour et al., 2014; Goldberg et al., 2016; Hascoet and Morlighem, 2018).

The adjoint of ice sheet model SICOPOLIS is largely generated automatically by the application of the freely available source-transformation tool OpenAD (Utke et al., 2008), developed at Argonne National Laboratory, University of Chicago, and Rice University; www.mcs.anl.gov/OpenAD. It is a flexible and modular tool that parses a given model written in Fortran to generate a Fortran version of the model's adjoint code.

The adjoint model of SICOPOLIS produced by OpenAD thus results in approximately 50k executable lines (represented in a much simplified schematic in Figure 1 by the composition of the blue, red, and black algorithmic steps), depending on C preprocessor (CPP) options enabled or disabled at compilation time (which include or exclude source code to be differentiated). An advantage of code exclusion at compile time (as opposed run-time selection) is that real or artificial code flow dependencies can be reduced (or avoided) in the code's analysis. By pairing SICOPOLIS with source-transformation tool OpenAD, the adjoint model of SICOPOLIS may be generated automatically, for a large variety of forward model configurations (including detailed choices of model domain, numerics, as well as control variables and QoI).

A number of algorithmic aspects of the code needed one-time editing or refactoring for OpenAD to be able to successfully parse the source code and provide correct adjoint code. For example, non-smooth functions – such as piecewise linear functions represented by IF-statements or absolute values – are inherently non-differentiable, sometimes required special treatment before the adjoint could be obtained by AD. Because of its importance in the development of a forward model that works properly within the framework of the AD tool, we have devoted a detailed description in Appendix B of the aspects of SICOPOLIS that required code refactoring in Appendix. Further technical details on how to set up, compile and run reference configurations are documented in a Quick-Start Manual (Logan et al., 2019).

## 3 Example application: Antarctic ice sheet volume sensitivities

Because SICOPOLIS is capable of simulating many different aspects of ice flow at the continental scale, we have designed a set of configurations focusing in each on particular aspects of the model, so that the resulting adjoint values and patterns may be more readily interpreted. Where we could have applied more complicated relationships in, for example, initialization in temperature, geothermal heat flux, or calving laws, we have opted instead for simplicity, as the exhaustive examination of such choices in simulation are left to future studies. The adjoint values are calculated for specific configurations of the original, unmodified forward code of SICOPOLIS.





### 3.1 Antarctic model configuration

We simulate Antarctica for 100 yr of model time with a 20 km horizontal resolution and 81 terrain-following vertical layers. The dynamic and thermodynamic time steps (which can be chosen to differ) were both set to 0.2 yr, as this was found to be the most stable value for the adjoint run. Land ice, floating ice, and ice streams are approximated by the SIA / SSA /
SStA formulations described in Bernales et al. (2017). Ice thickness evolves freely and without adjustment. Solutions to the SSA portion of SICOPOLIS are aided by invoking the external Library of Iterative Solvers (LIS, https://www.ssisc.org/lis/). Thermodynamics are formulated by the conventional enthalpy scheme (Section 2.1). We use Glen's flow law (8) with a stress exponent $n = 3$, a residual stress $\sigma_0 = 10^4 \, \mathrm{Pa}$, uniform flow enhancement factors $E = 5$ for grounded ice and $E = 1$ for floating ice, and a rate factor $A(T')$ as in Cuffey and Paterson (2010). Horizontal and vertical advection in the temperature and age
equations are discretized via a first-order upstream stencil of interpolated velocities and advection terms on the main grid, and topography gradients are evaluated with a fourth-order discretization. The ice temperature is initialized as a uniform value of $-10°\mathrm{C}$, as the goal of this exercise is proof-of-concept and not exhaustive examination of all aspects of the Antarctic Ice Sheet. For the same reason, a constant geothermal flux of $55 \, \mathrm{mW \, m^{-2}}$ is applied uniformly. Parameterization of the mean-annual and mean-January surface temperatures is according to Fortuin and Oerlemans (1990), and the applied surface temperature is held
constant throughout the simulation. Accumulation is applied at present-day levels throughout the simulation (Le Brocq et al., 2010; Arthern et al., 2006). The fraction of solid precipitation is a linear function of the monthly mean surface temperature according to Marsiat (1994). Surface ablation is parameterized by the positive degree day method, and rainfall is assumed to run off instantaneously. Floating ice is removed at calving fronts for thicknesses less than 30 m. The parameters for the basal sliding law (9) are chosen as $C_\mathrm{b} = 11.2 \, \mathrm{m \, yr^{-1} \, Pa^{-1}}$, $p = 3$ and $q = 2$. Basal melting under floating ice is set to $30 \, \mathrm{m \, water \, equiv. \, yr^{-1}}$
around the grounding zone (adjacent grounded and floating grid points) and zero elsewhere, for simplicity. Sea level is constant, and there is no special treatment of subglacial hydrology. The initial geometry is taken from the present day (Fretwell et al., 2013). There is no isostatic bedrock adjustment during the 100 yr simulation.

### 3.2 Results

The motivation for developing an adjoint of a numerical model stretches far beyond providing comprehensive sensitivity
experiments; often, an adjoint model is developed so that the sensitivities may be used in a constrained optimization problem to invert for interesting initial and boundary conditions, as well as the evolution of the state of the system. Here, however, we present the sensitivity of the volume of the Antarctic Ice Sheet with respect to several control variables as a proof of concept, rather than extending the work in the direction of optimization, which will be the subject of future studies. The purpose is to gain physical insight into the model's linear response characteristics and to ascertain correctness and interpretability of the adjoint.
The adjoint-derived sensitivities are compared to finite difference perturbations, either at single points, or over a patch of the domain that has been uniformly perturbed, to demonstrate that the adjoint model sufficiently approximates sensitivities derived via finite-differences. Those comparisons are shown in Table 1. Lastly, we present this work for the novelty of examining

| Variable (1) | Region (from Fig. 2A) (2) | $\frac{\delta J}{\delta \text{Variable}}$ (3) | $\frac{\Delta J}{\Delta \text{Variable}}$ (4) | $\Delta V_{\text{adj}}$ (5) | $\Delta V_{\text{fd}}$ (6) | % Deviation (7) |
|---|---|---|---|---|---|---|
| surface temperature | 1 | $-4.87 \times 10^7$ | $-4.67 \times 10^7$ | – | – | 4.43 |
| basal temperature | 1 | $-1.25 \times 10^8$ | $-1.15 \times 10^8$ | – | – | 8.74 |
| January precipitation | 2 | $7.50 \times 10^{16}$ | $8.53 \times 10^{16}$ | – | – | 12.09 |
| ice thickness | 3 | $3.62 \times 10^8$ | $3.68 \times 10^8$ | – | – | 1.40 |
| surface temperature | 4 | – | – | $-7.26 \times 10^3$ | $-1.45 \times 10^4$ | 49.85 |
| basal temperature | 4 | – | – | $-1.22 \times 10^7$ | $-2.88 \times 10^7$ | 57.52 |
| ice thickness | 4 | – | – | $1.12 \times 10^{11}$ | $1.32 \times 10^{11}$ | 15.01 |
| January precipitation | 4 | – | – | $5.89 \times 10^{18}$ | $5.54 \times 10^{18}$ | 6.25 |

**Table 1.** A sample of comparisons between adjoint-derived sensitivities and finite-difference-based sensitivities. All regions in column (2) refer to either selected points or the box from Fig. 1A. Columns (3) and (5) are adjoint-derived quantities, while columns (4) and (6) are derived via finite-difference perturbations, either to a single point in the domain or a patch, as shown in Fig. 2A. Validation of the adjoint model is attempted by comparing finite-differences (4) and adjoint values (3), in a % deviation metric (7). Column (7) is calculated as $\frac{|\text{col.}(4) - \text{col.}(3)|}{\text{col.}(4)} \times 100$. Ice thickness, surface, and basal temperatures compare well, with a % deviation of less than 10%. Summer precipitation has a higher disagreement, at 12 %. We also performed a test for the Greenland Ice Sheet, shown in more detail in Appendix A.

Antarctic-wide adjoint-generated sensitivity maps as, to the authors' knowledge, such a presentation has not been formally examined heretofore.

Figures 2 and 3 show, respectively, the raw and logarithm of the absolute value ($\log_{10} |\bullet|$) of the adjoint sensitivities. We have chosen to present the adjoint values in both ways so that the general pattern and sign of the adjoint values are readily apparent (Figure 2) as well as the order of magnitude of the adjoint values (Figure 3), which can vary widely across the Antarctic Ice Sheet depending on the control variable. Further, we have chosen the locations shown in Figure 2A so that the included dynamics and solvers invoked in the code can be tested in three different and important regions: location 1, the fast-moving Thwaites Glacier, which directly discharges into the Amundsen Sea Embayment; location 2, the middle of the Ross Ice Shelf; and location 3, Slessor Ice Stream, which feeds the Ronne-Filchner Ice Shelf. Testing the agreement between adjoint and finite-difference values at these locations offers a broad sense of the performance of the adjoint model in very different and important environments and dynamical regimes across the Antarctic Ice Sheet. The control variables we have selected to test involve either initial (ice thickness) or boundary conditions (summer precipitation, surface and basal temperature), and are independent inputs to either the conservation of mass (ice thickness and summer precipitation) or conservation of enthalpy (surface and basal temperature) equations.

The sensitivity of total Antarctic ice volume to the initial ice thickness compares well with the calculated finite-difference based value (Table 1, col. 7), differing only by about 1% at the fast-moving Slessor Ice Stream (2A, location 3). Thickness



**Figure 2.** Adjoint sensitivities, $\frac{\delta J}{\delta X_i}$, for the Antarctic Ice Sheet, where $X_i$ is the control variable shown. Control variables are the [A] initial ice thickness (units $\mathrm{m}^2$), [B] mean July precipitation (units $\mathrm{m}^2\,\mathrm{yr}$), [C] surface temperature (units $\mathrm{m}^3\,{}^\circ\mathrm{C}^{-1}$), and [D] basal temperature (units $\mathrm{m}^3\,{}^\circ\mathrm{C}^{-1}$). Locations in [A] numbered 1-4 are compared to finite-difference values in Table 1.

sensitivities are relatively uniform and positive across the ice sheet over the 100 yr simulation, except for a few outlet glaciers. A positive adjoint value in ice thickness indicates that a positive perturbation in ice thickness leads to a positive change in total volume, and vice versa. The Antarctic Ice Sheet is shown to have almost entirely positive adjoint values, as shown in Figure 2A, except for a few marginal outlet glaciers. These few outlet glaciers that display negative ice thickness adjoint sensitivities contrast with other areas of ice discharge, notably, the large floating ice shelves, which do not show any negative adjoint values. Figure 3A shows that the order of magnitude of this field of adjoint values is between $10^8$ and $10^9\,\mathrm{m}^2$, except for several very



**Figure 3.** Logarithms of absolute value of adjoint sensitivities, $\log|\frac{\delta J}{\delta X_i}|$, for the Antarctic Ice Sheet, where $X_i$ is the control variable shown. Control variables are the [A] initial ice thickness (units $\mathrm{m}^2$), [B] mean July precipitation (units $\mathrm{m}^2\,\mathrm{yr}$), [C] surface temperature (units $\mathrm{m}^3\,{}^\circ\mathrm{C}^{-1}$), and [D] basal temperature (units $\mathrm{m}^3\,{}^\circ\mathrm{C}^{-1}$).

sensitive outlet glaciers, including Thwaites and Pine Island Glaciers, glaciers in Marie Byrd, Oates, and Wilkes Lands, and Byrd Glacier.

The pattern of the January (austral summer) precipitation adjoint values largely mirrors that of the ice thickness, with several distinctions. The order of magnitude is much larger, ranging instead between $10^{15}$ and $10^{17}\,\mathrm{m}^2$ (figure 3B). Outlet glaciers in Marie Byrd, Oates, Wilkes, and Queen Mary Lands exhibit weaker sensitivities compared to the average Antarctic-wide summer precipitation sensitivities. Portions of floating ice shelves from Queen Maud eastward all the way to Queen Mary Land show the highest sensitivities overall to summer precipitation, while the larger ice shelves exhibit some of the lowest sensitivity


to summer precipitation across the entire continent, almost an order of magnitude lower than the floating ice fringing the coast between Queen Maud and Queen Mary Lands, from $10^{17}$ to $10^{16}\,\mathrm{m}^2$. Similar to the sensitivities to ice thickness, precipitation sensitivities are almost entirely positive, and the very lowest sensitivities are largely at the ice fronts (figure 3B). Table 1 shows less agreement in the January precipitation field calculated in the middle of the Ross Ice Shelf at point 2 (Figure 2A), approximately a 12% difference.

Sensitivities to surface and basal temperature (figure 2C and D) differ in pattern and sign from those of ice thickness and precipitation. The sensitivities of total ice sheet volume to surface and basal temperature are largely negative, with the most negative values at the margins of the ice sheet and approaching zero toward the interior. The order of magnitudes of the surface and basal temperature sensitivities (figure 3C and D) are comparable to each other, with maximum values of approximately $10^{10}\,\mathrm{m}^3\,{}^\circ\mathrm{C}^{-1}$. Over the 100 yr simulation, high sensitivities to surface and basal temperature at the margins extend inward toward the middle of the ice sheet following glacier drainage basins (figure 3C and D). There are two distinct differences between the sensitivities to surface and basal temperature seen in the adjoint fields. First, the highest sensitivities to basal temperature are higher than the highest sensitivities to surface temperature, indicating that the total ice sheet volume is in general more sensitive to changes in the applied basal temperature of the ice rather than at the surface in SICOPOLIS. Second, Figure 3D shows that the location of those most sensitive areas to changes in basal temperature are at the grounding lines of ice shelves and glaciers, while the most sensitive areas to changes in surface temperature are all across the surface of the floating portions of ice, with the sensitivity increasing (becoming more negative) toward the ice fronts. The adjoint values of surface and basal temperature compare well with finite difference based sensitivities, differing by about 4 and 8 %, respectively.

Table 1 also shows the results of a finite volume change calculation performed for the tile shown in Figure 2A, region 4. Each control variable shown in column 1 is perturbed, as in the single point location finite differences, by $\pm 5\%$ of the initial value of that field.

Columns 5 and 6 in Table 1 are computed in the following ways. $\Delta V_{\mathrm{adj}} = \int_{\Omega_4} \frac{\delta J}{\delta X} \delta X$ is the adjoint-derived volume change, where $\Omega_4$ is the domain of the tile in 2A and $X$ is the control variable. $\Delta V_{\mathrm{fd}} = \frac{V(X+\delta X) - V(X-\delta X)}{2\delta X}$ is the finite difference-derived volume change, where $X + \delta X$ is taken over the entire tile in region 4. In creating a sub-domain of Antarctica over which to calculate these finite volume changes, we selected an area of uniform sign. Areas with a great deal of sign variation might be more difficult to interpret since the adjoint values would tend to cancel each other.

The utility of this comparison is to convert the sensitivities into meaningful quantities that can be compared against each other, to assess for example, which control variable impacts the cost function the most, given a perturbation of expected magnitude, in addition to providing another metric by which we may measure the adequacy of the adjoint model of SICOPOLIS. The % difference between the $\Delta V_{\mathrm{adj}}$ and $\Delta V_{\mathrm{fd}}$ over the 100 yr time integration over tile 4 is largely higher than the point-wise measurements, ranging as high as 57% for volume changes due to basal temperature perturbations uniformly in tile 4. Summer precipitation compares well, however, with a 6% difference. Perhaps more interesting, the calculations in columns 5 and 6 suggest that overall, summer precipitation has the largest impact on total ice sheet volume, with an approximate volume change of $10^{18}\,\mathrm{m}^3$, compared against initial ice sheet thickness, surface, and basal temperature perturbations, which at the lowest resulted in a volume loss of $10^3$ or $10^4\,\mathrm{m}^3$ over 100 yr. This result helps to explain the largest relative differences between adjoint and





finite-difference sensitivities. These are large where the sensitivities are very small compared to the QoI, strongly suggesting that numerical noise plays an important role in either of these sensitivity calculations.

Lastly, the adjoint model of SICOPOLIS runs serially, and completed 100 yr of model run time in 20, 75, and 600 minutes of wall clock time on a Linux box (Intel Xeon CPU E5-2650 at 2.00 GHz) for resolutions of 64, 40, and 20 km. The results
shown in Figures 2 and 3 are for 20 km resolutions.

## 4   Discussion

The results presented here are not meant to be exhaustive, but rather present initial adjoint sensitivity applications of the newly AD-enabled SICOPOLIS model. They underscore the interpretable nature of adjoint-derived sensitivity fields and are presented as a proof of concept for further investigation. We leave an exhaustive study of sensitivities to different control variables in
SICOPOLIS to future work, as here we only wish to examine a few important dynamic and thermodynamic controls and assess the validity of the adjoint model. As a measure of the adjoint model's correctness, we compared gradients obtained from the adjoint model and computed via finite differences perturbations. Adjoint values compared acceptably against finite differences for ice thickness, surface, and basal temperatures, with less than 10% deviation. Austral summer precipitation adjoint values saw a larger disagreement with finite differences, of up to 12%. Part of the higher discord may be due to the
fact that the cost values (total Antarctic Ice Sheet volume) are very large, emphasizing numerical noise for sensitivity fields that are very small. Ice sheet volume changes calculated by the adjoint model and finite differences disagree more, although the largest discrepancy occurred with the smallest overall volumes calculated (both surface and basal temperature) and are thus likely, again, to be affected by numerical noise arising in the calculation. Control variables related to the conservation of mass equation provided the best agreement across measured metrics (ice thickness for point-wise sensitivities and precipitation for
finite volume calculations). This is readily explained by the primarily linear nature of precipitation changes (seen as a volume flux) in changing total ice volume.

The general similarity between ice thickness and precipitation adjoint sensitivities (Figure 2A and B) is reassuring, as ice thickness and precipitation are both terms in the conservation of mass equation, and as such are algorithmically linked. In particular, we might expect adjoint sensitivities to be linearly related. Similarly, we might expect the sensitivity to summer
precipitation to be much larger than the sensitivity to initial ice thickness, as a perturbation in an initial condition is applied only once at model initialization, while a (constant in time) perturbation in the surface boundary condition is iterated for every time step throughout the 100 yr of model run time. One of the largest obvious differences between Figures 2A and B is the appearance of high sensitivity on the floating portions of ice off the coasts between Queen Maud Land eastward to Queen Mary Land. Assuming that the "direct" linear effect of an increased precipitation (volume flux) has the same effect everywhere on
increasing the ice sheet volume, the difference between thin floating ice shelves and ice sheet interior may be explained by the dynamical effect of increased ice shelf buttressing (i.e., reduced mass flux through the grounding line) as a consequence of ice shelf mass accumulation.





In a related way, the overall similarity of the surface and basal temperature sensitivities is reassuring as both of these are components in the same conservation of energy equation. Both fields of sensitivities delineate the drainage basins of glaciers and ice shelves, with very small sensitivity in the center of the ice sheet that increases by orders of magnitude toward the coasts. The surface temperature sensitivities more uniformly affect total ice volume over the ice shelves, while the basal temperature

sensitivities indicate that positive perturbations in basal temperature at the grounding lines of glaciers and ice shelves have a larger effect on total ice volume, and that when compared with each other, variations in basal temperature are more powerfully felt across the Antarctic Ice Sheet. This seems to indicate that changes in ocean temperature at the grounding lines around Antarctica have much more potential to do lasting damage to the volume of the ice sheet than temperature changes in the atmosphere. However, this conclusion must be tempered by the fact that our current simulation of the surface of ice does not

account for melt water ponding and induced catastrophic failure, as has been observed in the past at the Larsen B Ice Shelf, for example (Glasser and Scambos, 2008). Thus in the context of our finite volume calculations performed in Table 1, columns 5 and 6, while the effect of summer precipitation applied uniformly every 0.2 yr during the 100 yr simulation to the tile in sub-domain 4 from Figure 2A results in volume change that dwarfs the effects of the thermal controls applied in the same region, the same result may not hold in different regions with more complicated formulations for surface or basal melting, or

using the more sophisticated calving relations available within the main trunk of SICOPOLIS.

Algorithmic differentiation relies on algorithms being differentiable, line by line, in a code. Numerical disagreement can accumulate for even simple reasons, such as the use of piece-wise linear functions represented algorithmically by IF-statements (see Appendix B for a larger discussion on unstructured code and non-smoothness introduces error in adjoint codes developed by AD).

## 20  5  Conclusions

This work presents a new capability of the ice sheet model SICOPOLIS to enable flexible adjoint code generation using the open-source AD tool OpenAD. The flexibility is afforded by allowing a wide range of choices of model domains, numerical algorithms chosen for specific configurations, as well as control variables and quantities of interest (cost functions) defined, when generating the adjoint code. We demonstrate the utility, correctness and interpretability of adjoint-derived sensitivity

maps for Antarctic-wide simulations, with the total volume of the Antarctic Ice Sheet chosen as quantity of interest, and subject to sensitivities in initial and boundary conditions over a 100 yr simulation from present day. Examining and assessing the information contained in such sensitivity maps, which are formally gradients of scalar-valued functions with respect to model inputs, is a useful and natural first step in the use of these sensitivities in gradient-based optimization problems, which will be the subject of future work.

One suggested outcome of the sensitivity analysis is that, as a controlling variable, mean monthly applied summer precipitation influences the total integrated Antarctic Ice Sheet volume more than the initial ice geometry, surface, or basal temperatures do. Another hypothesized (and perhaps unsurprising) relationship derives from a comparison between the surface and basal ice





temperatures: that changes in basal temperature, particularly at grounding lines affect total ice volume much more than those in surface temperature.

Much remains to be learned and further examined in the context of this model, as well as the degree to which results may be applicable to other models. Our results are specific for a given configuration of SICOPOLIS, with emphasis placed on the initial use of simple parameterizations for (often) the most interesting aspects of ice flow, including how basal melting or firn compaction are represented (both processes would be affected by the control variables chosen here). Our metrics of model validity evaluated point-wise show that the adjoint model is mostly accurate to within 10% compared to sensitivities obtained via the finite difference method. One likely reason the for larger disagreements in some of the calculated metrics may be due to the regimes of very weak sensitivities, in which case numerical noise becomes a leading factor in the inferred differences.

Another cost function may be formulated as a model-data misfit based on, for example, the modeled versus observed spatio-temporal ice elevation change. Additionally, over-reliance on inherently non-differentiable piece-wise linear functions for important aspects of surface mass balance terms may introduce discrepancies that could be minimized with the use of smoother functions. These are valid and important aspects of code that are not easily addressed. We have described in some detail code-refactorization steps that was required for SICOPOLIS to comply with code parsing and analysis steps undertaken by OpenAD in Appendix B. Many of the issues described in the Appendix are frequently encountered when subjecting legacy code to AD, or when considering the development of new code that should be subjected to AD. The Appendix thus provides insights for coding best-practices in the context of AD beyond the application to SICOPOLIS.

As glaciologists strive to make ever-more confident projections in the future behavior of ice sheets, tools that rigorously determine the relationship between often poorly known input parameters and important model outputs are increasingly needed. SICOPOLIS-AD is one such tool that is freely available to the cryosphere community (Logan et al., 2019) and, as demonstrated here, can help elucidate relationships between model inputs and outputs that were previously unknown or untested.

*Code availability.* SICOPOLIS is free and open-source software, available through a persistent Subversion repository that is hosted by the FusionForge system AWIForge of the Alfred Wegener Institute for Polar and Marine Research (AWI) in Bremerhaven, Germany (https://swrepo1.awi.de/). Detailed instructions for obtaining and compiling the code are at http://www.sicopolis.net. The adjoint genera-tion capability of SICOPOLIS is a part of the main trunk of the current developmental version (5-dev). The development and tests were performed using SICOPOLIS v5-dev (revision 1414), tagged as SICOPOLIS-AD v1. It can be specifically downloaded at https://swrepo1.awi.de/svn/sicopolis/tags/ad-v1. The AD tool used to generate adjoint source code is OpenAD. OpenAD, release tag "SICOPOLIS-AD v1", can be downloaded at https://doi.org/10.5281/zenodo.3361744. Detailed instructions on how to download and build the tool are at https://www.mcs.anl.gov/OpenAD/. Technical details on how to set up, compile and run reference configurations of SICOPOLIS-AD are documented in a Quick-Start Manual (Logan et al., 2019).





**Appendix A:  Greenland ice volume sensitivities**

Here we present the results of a 100 yr sensitivity study of Greenland ice sheet volume to basal ice temperature. This is added as an Appendix as Greenland sensitivities have been produced previously by Heimbach and Bugnion (2009), albeit with a different AD tool, Transformation of Algorithms in Fortran (TAF; Giering et al., 2005), for a version of the SICOPOLIS model

that is more than a decade old, and which was not maintained as part of SICOPOLIS' main development trunk. Since then, SICOPOLIS has been updated to include a more state-of-the-art representation of thermodynamics via the enthalpy method, and work here represents an advance on what was presented before.

The forward simulation of Greenland is configured in much the same way as for Antarctica, with an emphasis on simplicity for proof-of-concept. Unless otherwise stated below, choices of numerical schemes, physical parameterizations, and forcings

approaches are the same. We simulate Greenland for 100 yr from present day at a 10 km horizontal resolution with 81 terrain-following vertical layers. The dynamic and thermodynamic time steps again take the same value of 0.5 yr. The dynamics now are only SIA, as we have restricted our simulation to grounded ice. The thermodynamic formulation is again via the conventional enthalpy method, with ice initialized at a constant temperature of $-10°C$. The constitutive law and physical parameters are exactly the same as in the Antarctic case, including the flow enhancement factor, geothermal flux, and all

parameters for the sliding law. The ice initial geometry is from Bamber et al. (2013). Surface temperature is from Ritz (1997) and is held constant throughout. The monthly precipitation fields are created with the regional energy and moisture balance model REMBO using the setup described in Robinson et al. (2010) taken on the grid provided by Bamber et al. (2001). The temperature and humidity boundary conditions are from Uppala et al. (2005). These monthly climatological fields are averaged over 1958-2001 and applied as lateral boundary conditions to the REMBO.

Figure A1 shows the total Greenland Ice Sheet volume sensitivity to initial condition of ice thickness, and boundary conditions July (boreal summer) precipitation, surface, and basal temperatures. Table A1 shows that, in general, the Greenland simulation performs much better than the Antarctic simulation, with all of the % deviations between adjoint values and finite

**Table A1.** Comparison between adjoint-derived (column 3) and finite difference derived (column 4) sensitivities for Greenland ice volume as QoI. All regions in column (2) refer to points from figure A1A. Column (5) is a % deviation metric, which is calculated as $\frac{|\text{col.}(4)-\text{col.}(3)|}{\text{col.}(4)} \times 100$.

| Variable | Region (from Fig. A1A) | $\frac{\delta J}{\delta \text{Variable}}$ | $\frac{\Delta J}{\Delta \text{Variable}}$ | % Deviation |
|---|---|---|---|---|
| (1) | (2) | (3) | (4) | (5) |
| July precipitation | 1 | $2.72 \times 10^{16}$ | $2.72 \times 10^{16}$ | $3.37 \times 10^{-3}$ |
| surface temperature | 2 | $-5.61 \times 10^{4}$ | $-5.57 \times 10^{4}$ | $7.18 \times 10^{-1}$ |
| basal temperature | 2 | $-3.80 \times 10^{6}$ | $-3.80 \times 10^{6}$ | $8.20 \times 10^{-3}$ |
| ice thickness | 3 | $4.86 \times 10^{2}$ | $4.89 \times 10^{2}$ | $5.04 \times 10^{-1}$ |

**Geoscientific Model Development Discussions**

**Figure A1.** Adjoint sensitivities, $\frac{\delta J}{\delta X_i}$, for the Greenland Ice Sheet, where $i$ is the control variable shown. Control variables are the [A] initial ice thickness (units $\mathrm{m}^2$), [B] mean July precipitation (units $\mathrm{m}^2\,\mathrm{yr}$), [C] surface temperature (units $\mathrm{m}^3\,{}^{\circ}\mathrm{C}^{-1}$), and [D] basal temperature (units $\mathrm{m}^3\,{}^{\circ}\mathrm{C}^{-1}$). Locations in [A] numbered 1-3 are compared to finite-difference values in table A1

difference based gradients less than 1%. We attribute this to the lack of SSA dynamics involved and the accompanying use of an external solver library.

Interestingly, whereas in Antarctica the ice thickness sensitivities were almost entirely positive, substantial portions of the Greenland Ice Sheet loses volume when perturbed positively in ice thickness, a phenomenon previously inferred by Heimbach and Bugnion (2009). This could be due to dynamic draw down of glaciers that experience a sudden increase in driving stress due to the increase in ice thickness. The increase in driving stress leads to increases in velocity which, when subjected to the land-ice-only mask for the SIA dynamics used in this setup of Greenland, results in the immediate cutoff of ice.





Precipitation, as in the case of Antarctica, is almost entirely positive, and again, dwarfs the other control variables tested here by many orders of magnitude. The overall larger magnitude of basal temperature sensitivities compared to surface temperatures is consistent with the Antarctic simulation. Completion of Greenland serial qsimulations on a Linux box (Intel Xeon CPU E5-2650 at 2.00 GHz) took 5, 10, and 140 minutes for horizontal resolutions of 40, 20, and 10 km, respectively. The results shown here are for 10 km resolution.

## Appendix B: Modifying SICOPOLIS

We made several modifications to SICOPOLIS to enable source transformation and differentiation via OpenAD. The changes that were made enabled efficient AD in some cases and overcome some limitations of the AD tool used in others. The modifications are guarded by C preprocessor (CPP) directive **ALLOW_OPENAD** and do not affect the original behavior of SICOPOLIS in any way. Below, we discuss the noteworthy changes.

- **Data Types:** SICOPOLIS determines the number of bits for its data types at runtime through the call `selected_int_kind ()` and `kind()`. Because OpenAD requires full knowledge of the types for static analysis, it does not support this behavior. We determine, therefore, the number of bits-per-type separately for the machine being used and specify the value directly in the code.

```
#ifndef ALLOW_OPENAD
integer, parameter :: i1b = selected_int_kind(2)    !< 1−byte integers
integer, parameter :: i2b = selected_int_kind(4)    !< 2−byte integers
integer, parameter :: i4b = selected_int_kind(9)    !< 4−byte integers
integer, parameter :: sp  = kind(1.0)               !< Single−precision reals
integer, parameter :: dp  = kind(1.0d0)             !< Double−precision reals
#else
integer, parameter :: i1b = 4
integer, parameter :: i2b = 4
integer, parameter :: i4b = 4
integer, parameter :: sp  = 4
integer, parameter :: dp  = 8
#endif
```

- **Unstructured code** The adjoint model of OpenAD reverses the control flow of the original code, including those of loops. It uses the following criteria to evaluate whether the loops are *simple*.

    1. loop variables are not updated within the loop,

    2. the loop condition does not use `.ne.`,

    3. the loop condition's left-hand side consists only of the loop variable,

    4. the stride in the update expression is fixed,




5. the stride is the right-hand side of the top level `.+` or `.-` operator,

6. the loop body contains no index expression with variables that are modified within the loop body

SICOPOLIS contained several cases of statements injected to break out of loops which cause them not to be simple. To differentiate non-simple loops correctly, OpenAD stores which array indices are actually used per loop iteration. This approach causes significant memory usage and performance loss. Therefore, we removed the `exit` statements by rewriting the loop body to include a conditional statement that executes the loop only when the original loop would not exit. Restricting code to comply with "simple loops" is common in models subject to AD, and is good coding practice in general, as it supports compiler optimization of loops.

```fortran
#ifndef ALLOW_OPENAD /* Normal */

do kc=1, KCMAX-1
   if (omega_c_neu(kc,j,i) > eps_omega) then
      kc_cts_neu(j,i) = kc
   else
      exit
   end if
end do

#else /* OpenAD */

kcdone = .false.

do kc=1, KCMAX-1
   if (kcdone.eqv..false.) then
      if (omega_c_neu(kc,j,i) > eps_omega) then
         kc_cts_neu(j,i) = kc
      else
         kcdone = .true.
      end if
   end if
end do

#endif /* Normal vs. OpenAD */
```

– **Nonsmoothness** Non-smoothness in the underlying mathematics of a model can be caused by the use of the absolute value, ceiling, and floor functions. Non-smooth models can be non-differentiable at a few or many points of the input space. Techniques such as piecewise linear differentiation and the *absnormal* form have been studied to differentiate non-smooth applications (Streubel et al., 2014). While SICOPOLIS employs all three functions, they are either used to index into lookup tables or used in portions not differentiated by OpenAD. Because OpenAD does not, however, include





`abs`, `ceiling`, and `floor` as *functions* within its intrinsic library, we created custom *subroutines* of these functions to be differentiated.

```
#ifndef ALLOW_OPENAD /* Normal */
n_filter = ceiling(2.0_dp*sigma_filter)
#else /* OpenAD */
call myceiling(2.0_dp*sigma_filter, n_filter)
#endif /* Normal vs. OpenAD */
```

– **sqrt function** The derivative of $\sqrt{x}$ is $\frac{1}{\sqrt{x}}$. When $x = 0.0$, the result is a *kink* in the adjoint model and the appearance of `NaN` in the adjoint computation. The intended behavior of the adjoint model is to treat the derivatives as 0.0. Therefore, wherever the function `sqrt()` appears in SICOPOLIS, we use a conditional to check if the input to `sqrt()` is `0.0` and in those cases we use `0.0` instead of calling `sqrt()`.

```
do i=0, IMAX
do j=0, JMAX

#ifndef ALLOW_OPENAD /* Normal */

   tau_b(j,i) = p_b(j,i)*sqrt(dzs_dxi_g(j,i)**2+dzs_deta_g(j,i)**2)

#else /* OpenAD: guarding against non-differentiable sqrt(0) */

   if ((dzs_dxi_g(j,i)**2+dzs_deta_g(j,i)**2) > 0) then
      tau_b(j,i) = p_b(j,i)*sqrt(dzs_dxi_g(j,i)**2+dzs_deta_g(j,i)**2)
   else
      tau_b(j,i) = 0.0_dp
   end if

#endif /* Normal vs. OpenAD */
end do
end do
```

– **Array Declaration and Array Assignments** SICOPOLIS uses dynamic memory allocation for some arrays in the code. Because the handling of dynamic memory and pointers by source transformation AD tools such as OpenAD remains a topic of active research we replaced the dynamic allocation with static allocation.

```
#ifndef ALLOW_OPENAD /* Normal */

real(dp), allocatable, dimension(:,:) :: f_0

#else /* OpenAD */
```



```
real(dp), dimension(-JMAX:2*JMAX,-IMAX:2*IMAX) :: f_0

#endif /* Normal vs. OpenAD */
```

SICOPOLIS uses constructs such as `where` and `elsewhere` to elegantly assign values to array elements. Because OpenAD does not support these constructs, we rewrote them using loops and **if** statements.

```
#ifndef ALLOW_OPENAD /* Normal */

  where ( (maske == 0_i2b).and.(H < rhosw_rho_ratio*H_sea) )
     calv_uw_ice = calv_uw_coeff * H**r1_calv_uw * H_sea**r2_calv_uw
  elsewhere
     calv_uw_ice = 0.0_dp
  end where

#else /* OpenAD */

  do i=0, IMAX
  do j=0, JMAX
     if ( (maske(j,i) == 0_i2b) .and. (H(j,i) < rhosw_rho_ratio*H_sea(j,i)) ) then
        calv_uw_ice(j,i) = calv_uw_coeff * H(j,i)**r1_calv_uw * H_sea(j,i)**r2_calv_uw
     else
        calv_uw_ice(j,i) = 0.0_dp
     end if
  end do
  end do

#endif /* Normal vs. OpenAD */
```

– **Intent of variables** SICOPOLIS passes the indices of two and three dimensional arrays as arguments with `intent(in)` to subroutines that act upon particular portions of the array. When these variables are not the type of *active* variable (usually `real(dp)`), their declarations must be changed to `intent(inout)`. For variables that are *active* OpenAD changes the intent automatically.

```
#ifndef ALLOW_OPENAD /* Normal */
integer(i4b), intent(in) :: ii
#else /* OpenAD */
integer(i4b), intent(inout) :: ii
#endif /* Normal vs. OpenAD */
```

– **Solvers** SICOPOLIS employs an array of solvers depending on the domain (e.g., Greenland versus Antarctica) or physics chosen by the user: a successive over relaxation (SOR) solver, a tridiagonal solver, and (for Antarctic domains) the library



of iterative solvers (LIS) for computing a system of linear equations:

$$A \cdot x = b \rightarrow x := \mathrm{solve}(A, b)$$

To differentiate the above formulation efficiently, an AD tool must not naively differentiate through the solver code. OpenAD uses its **template** mechanism instead to encode the formulation below (Giles, 2008) to compute the adjoints $\bar{A}$ and $\bar{b}$ from $\bar{x}$ using the original solver call.

$$A^T \cdot \bar{b} = \bar{x} \rightarrow \bar{b} := \mathrm{solve}(A^T, \bar{x})$$
$$\bar{A} := -x^T \cdot \bar{b}$$

When SICOPOLIS uses the SOR solver for a system of linear equations where the matrix storage is in compressed sparse row (CSR) format, arrays are represented by `lgs_a_value` (values), `lgs_a_index` (indices), and `lgs_a_ptr` (pointers). While the symbolic differentiation of the solver can be handled as above, the formation of the CSR representation requires us to change the type of the indices into `real(dp)` so that the indices are stored in the forward sweep for use in the reverse sweep.

```
#ifndef ALLOW_OPENAD /* Normal */
! ...
integer(i4b), allocatable, dimension(:) :: lgs_a_index
! ...
#else /* OpenAD */
! ...
real(dp),            dimension(n_sprs) :: lgs_a_index
! ...
#endif /* Normal vs. OpenAD */
```

– **Checkpointing** For adjoint models, the memory requirement to compute the adjoint information is proportional to the operation count of the model being differentiated. We found that the memory requirements of the adjoint model of SICOPOLIS for even small number of timesteps will quickly exceed the available memory of most machines. Therefore, we implemented a binomial checkpointing scheme using the library `revolve` (Griewank and Walther, 2000). This approach uses recomputation of timesteps in the original model to reduce the memory requirements of the adjoint model.

*Author contributions.* LCL, SHKN, and PH developed the adjoint code of SICOPOLIS; RG originally developed SICOPOLIS, provided insight to model results, and helped host the freely available version of the code.





*Competing interests.* There are no competing interests present.

*Acknowledgements.* This work was funded in part by the U.S. Department of Energy, Office of Science, under contracts DE-AC02-06CH11357 and SC0008060 (PISCEES), and the National Science Foundation, grant #1750035. Ralf Greve was supported by Japan Society for the Promotion of Science (JSPS) KAKENHI grant numbers JP16H02224, JP17H06104 and JP17H06323, and by the Japanese Ministry of Education, Culture, Sports, Science and Technology (MEXT) through the Arctic Challenge for Sustainability (ArCS) project.



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

The submitted manuscript has been created by UChicago Argonne, LLC, Operator of Argonne National Laboratory ('Argonne'). Argonne, a U.S. Department of Energy Office of Science laboratory, is operated under Contract No. DE-AC02-06CH11357. The U.S. Government retains for itself, and others acting on its behalf, a paid-up nonexclusive, irrevocable worldwide license in said article to reproduce, prepare derivative works, distribute copies to the public, and perform publicly and display publicly, by or on behalf of the Government. The Department of Energy will provide public access to these results of federally sponsored research in accordance with the DOE Public Access Plan. http://energy.gov/downloads/doe-public-access-plan.