# Peer review of "SICOPOLIS-AD v1: an open-source adjoint modeling framework for ice sheet simulation enabled by the algorithmic differentiation tool OpenAD"

_Geoscientific Model Development, 2019_

## Referee Comment (RC1) · Laurent Hascoet (Referee) · 3 Sep 2019

General comments:

This article describes a new development in the SICOPOLIS glaciology simulation code, to introduce sensitivity/adjoint/gradient computations. The article describes why adjoint capability is a significant improvement to a simulation code, allowing for sensitivity studies and solution to inverse problems and parameter estimation. The article discusses these new possibilities specifically for glaciology. This new adjoint capability was introduced through the use of an Algorithmic Differentiation tool: OpenAD. The article describes the amount of work that this AD tool required, and the amount of work

that it saved. The article also points to a few difficulties where AD tools still require the help of the end-user. Global performance of the adjoint-enabled code is described shortly. The article gives an in-depth discussion and interpretation of the obtained gradients for the glaciology and climate specialist, and points at further exploitation of this adjoint capability as further work. The article also provides some discussion about some observed deviation in the computed gradients.

Please note that I am not able to comment on the glaciology-specific parts of the text, although their general music seems completely reasonable.

The article is well structured and well written. It is easy to read, although some parts are obviously directed at true specialists of glaciology.

Like I write in the specific comments, I am slighty worried by the deviation observed for some gradient values, between adjoint and Divided Differences. I am only partly convinced by the explanation about numerical noise. The text also evokes the cases of non-smoothness of the implemented function. Could this be part of the explanation? I think this part of the discussion might be developed a bit, as some readers may really take it as an argument against AD.

I recommend publication of this article. It describes a solid work on an important code, it is useful as a clear example of what AD adjoints can do, it promotes AD towards the glaciology community, and it seeds for further work.

Specific comments:

P3, L12 : True, the adjoint propagation runs backward in time. More generally it runs backward the original simulation order (which happens here to be forward in time). Maybe it would be useful to stress that?

P7, L1 : Rather than "can be conceived as ...", I would advocate writing that "as soon as a numerical model is implemented as a code, it is in fact translated as ..."

P9, L15 : Are the preprocessor options used (to exclude or include arts) at "compilation time" or at "differentiation time" ? I take it that you mean "differentiation time". Does this imply then that there will be one particular adjoint model of SICOPOLIS for each model configuration. If so, your text presents this as an advantage but you understand some people might consider this as a drawback, not having a unique adjoint SICOPOLIS source at hand. (Here I'm playing the devil's advocate, as I think any source-transformation AD tool will face the same drawback)

P10, L3 : Does this raise the question about why, in the adjoint, some time steps are more stable than others? In other words, why can't one take the same time step sizess than in the forward simulation. That can be an interesting question for a Numerical Analysis specialist (not me...)

P10, L31: I'd replace "sufficently approximates" with "is sufficiently consistent with", because I tend to think that it is divided differences that is an approximation of the other.

P11, Table 1: I would swap rows 7 and 8 for consistency with rows 3 and 4. Deviations on rows 5 to 8 seem surprisingly high. Are they discussed in the text ? I read in P14 L20 that the finite difference chosen is around 5%, which can explain the high deviation. And yes, the explanation in P15 L1 may be right. But does the deviation decrease when the divided difference is smaller e.g. 0.5% instead of 5% ? As it is, a deviation of 57% is still worrying.

P15 L3: The question that comes immediately is how do these times compare with the primal simulation ? It might also be appropriate to describe the checkpointing scheme used in this experiment, on time-stepping: is it multi-level, or binomial, how many checkpoints are used, how many duplicated forward steps. Are these questions left for future work? Oh, I see it is in the appendix, P24 L24. Could you just, in the main text, point out that the appendix mentions that ?

P17 L17: The question of best practices makes me think that you may want to cite the Utke-Hascoet paper on that "Programming language features, usage patterns,..."

(OMS 2016)

Technical corrections:

P2, L25 : "construed"

P4, L13 : "more confident projections" -> "more faithful" ?

P5, L30 : "substantive ... to" -> "substantial ... over" ?

P6, L1: "ever" or "even" ?

P8, L15: why comma after warmer?

P9, L25: "appendix" is repeated

P10, L13: "instantaneously" ? "instantly"

P13, L3: Is it January? Figure 3 caption writes July. Or did I miss something?

P16, L18: "introduces" or "-introduced" ?
* * *

---

## Referee Comment (RC2) · Lizz Ultee (Referee) · 4 Oct 2019

**General comments:**

The manuscript by Logan et al. describes the generation of an adjoint code by algorithmic differentiation (AD) for the ice sheet model SICOPOLIS. The authors give a clear explanation of the motivation for adjoint modelling and for the use of AD in generating adjoint code, and they produce and interpret simple example applications for both Antarctica and Greenland.

I thank the authors for producing a well written and easy to follow manuscript. As a

non-expert in adjoint modelling, I found P3 an excellent description of its context and capabilities. I found the demonstration sensitivity analysis in Figures 2-3 interesting as well.

I am not the best person to comment on validation of the adjoint vs. finite-difference code, but I found the authors' explanations generally supportive of their conclusions. Perhaps a bit more data (e.g. absolute value of the QoI) could be given to support the comment about numerical noise producing high misfit in Table 1, Columns 5 and 6.

I agree with the authors that understanding uncertain input variables and models' sensitivity to them is important for contextualizing ice sheet/sea level projections. However, I disagree with the framing of the first paragraph of the introduction, namely that (A) effective adaptation to/mitigation of sea level rise relies on reducing uncertainty in projections and (B) development of more sophisticated ice sheet models will help reduce uncertainty.

Regarding (A): The social-science literature of climate adaptation discusses assorted factors that affect adaptive capacity, many of which have little to do with the state of the science. If the authors are interested, they could refer to e.g. Lemos and Rood 2010 (WIREs Climate Change) for a discussion of the "uncertainty fallacy" in climate science.

Regarding (B): It is intuitive that improved understanding of ice sheet dynamics will help us produce models that give more physically-consistent ("predictable") results. But physical consistency does not always translate to less uncertainty. For example, models that include "tipping point" dynamics (or hysteresis and multiple steady states) are arguably more sophisticated than those that do not, yet future projections over a range of climate scenarios may show a wider, not narrower, range when tipping point dynamics are included. Initial efforts to improve model sophistication by including newly-understood or newly-proposed processes can also increase uncertainty in terms of inter-model or inter-scenario spread. A notable example is the widening of 21st-22nd
century sea level projections shown in DeConto Pollard 2016 when the dynamics of marine ice cliff collapse and hydrofracture were included.

In a revised version, I suggest the authors strengthen the framing of the first paragraph to focus on the need for context and improved understanding, rather than leaning on the uncertainty angle.

The revisions I suggest to the introduction and in the specific comments below are relatively minor and should not impede publication. I imagine that many ice-sheet modellers will be interested in what the authors have shown here, and I look forward to reading follow-up studies using SICOPOLIS-AD.

**Specific comments:**

Figures 2 and 3 - both figure captions state that the [B] subplots illustrate sensitivities to July temperature. The text on P13,L3, Table 1, and later discussion refers to January or "summer" precipitation. Is there a mistake in the figure captions, or do the figures depict something not discussed in the text?

P4,L11 - "A model that can..." i.e. a forward model that can achieve the state deemed optimal by the Lagrange multiplier method? Does "reproduce the optimal behavior" refer to a model-vs-model or a model-vs-observation comparison?

P5,L10 - Is the adjoint code acceptable if the finite-difference-derived sensitivities approximate the adjoint-derived sensitivities, or is it the other way round? Intuitively I would expect that we accept the adjoint code if the sensitivities it produces approximate those derived by finite difference; that is, the finite-difference sensitivities are the "standard" against which the adjoint code is judged.

P5, Eq 3 - Given the explicit mention of "tolerance" in line 10, I might write the right-hand side of this equation with $= \frac{\partial \mathcal{J}}{\partial x} + \delta$, where $\delta$ is the accepted tolerance.

P5,L18 - This is a dense list of references without much discussion. Given that this

manuscript focuses on an ice-sheet application, the study might be well-served by adding another paragraph to discuss specific distinctions among these past efforts in adjoint modelling of ice sheets and any notable contrasts with the present study.

P10,L2-5 - What is the initial geometry? It is a bit unclear whether the experiment is 100 yr near equilibrium, a 100-yr spin-up, or something else. I don't know that it matters for the adjoint process, but it would be nice to have some more clarity.

P13, Fig 3 - I understand that the point of showing the logarithm of the absolute value of the sensitivities is so that the reader can compare their order of magnitude, both across parameters and within-parameter spatial variability. Is there a reason that each subplot uses a slightly different colormap? Could one colormap be applied to all subplots to facilitate intercomparison?

P17,L4-6 - Is there a use for adjoint modelling in distinguishing between different possible parameterizations of these processes?

**Technical comments:**

P2,L23 - For balance, an example of a quantity that "parameterize[s] subgrid-scale processes or empirical constitutive laws" would be helpful. Perhaps iceberg calving or the routing of surface/basal meltwater would be an appropriate example to include?

P2,L30 - "key quantities of interest that represent integrated quantities of an ice sheet" -> "ice-sheet-integrated quantities of interest"?

P9,L22 - "...inherently non-differentiable, sometimes required..." is a comma splice. Replace comma by "and"?

P16,L11 - "Thus ... within the main trunk of SICOPOLIS" is a very long sentence and I had to read it multiple times to understand. Consider streamlining.

P16,L18 - The phrase in the parentheses is hard to parse and might be a run-on. The authors might consider replacing the remarks in parentheses with another full sentence

or two to flesh out the thought.

P20,L1 - "Precipitation" or the "*sensitivity* to precipitation" is almost entirely positive?

P20, L3 - Typo "qsimulations"

―――――――――――――――――

---

## Author Comment (AC1) · 1 Dec 2019

**We would like to thank the reviewer for his careful read of the manuscript and constructive comments. In the following, we respond comment-by-comment (our replies are in bold-face).**

General comments:
This article describes a new development in the SICOPOLIS glaciology simulation code, to introduce sensitivity/adjoint/gradient computations. The article describes why adjoint capability is a significant improvement to a simulation code, allowing for sensitivity studies and solution to inverse problems and parameter estimation. The article discusses these new possibilities specifically for glaciology. This new adjoint capability was introduced through the use of an Algorithmic Differentiation tool: OpenAD. The article describes the amount of work that this AD tool required, and the amount of work that it saved. The article also points to a few difficulties where AD tools still require the help of the end-user. Global performance of the adjoint-enabled code is described shortly. The article gives an in-depth discussion and interpretation of the obtained gradients for the glaciology and climate specialist, and points at further exploitation of this adjoint capability as further work. The article also provides some discussion about some observed deviation in the computed gradients.

Please note that I am not able to comment on the glaciology-specific parts of the text, although their general music seems completely reasonable.
The article is well structured and well written. It is easy to read, although some parts are obviously directed at true specialists of glaciology.

Like I write in the specific comments, I am slighty worried by the deviation observed for some gradient values, between adjoint and Divided Differences. I am only partly convinced by the explanation about numerical noise. The text also evokes the cases of non-smoothness of the implemented function. Could this be part of the explanation? I think this part of the discussion might be developed a bit, as some readers may really take it as an argument against AD.

I recommend publication of this article. It describes a solid work on an important code, it is useful as a clear example of what AD adjoints can do, it promotes AD towards the glaciology community, and it seeds for further work.

Specific comments:
P3, L12 : True, the adjoint propagation runs backward in time. More generally it runs backward the original simulation order (which happens here to be forward in time). Maybe it would be useful to stress that?

**This is an excellent point and we have clarified this in the revision.**

P7, L1 : Rather than "can be conceived as ...", I would advocate writing that "as soon as a numerical model is implemented as a code, it is in fact translated as ..."
**We agree and rewrote this.**

*P9, L15 : Are the preprocessor options used (to exclude or include arts) at "compilation time" or at "differentiation time" ? I take it that you mean "differentiation time". Does this imply then that there will be one particular adjoint model of SICOPOLIS for each model configuration. If so, your text presents this as an advantage but you understand some people might consider this as a drawback, not having a unique adjoint SICOPOLIS source at hand. (Here I'm playing the devil's advocate, as I think any source-transformation AD tool will face the same drawback)*

**We appreciate the reviewer's concern, and the notion that some readers will regard this as a drawback. In the revised manuscript we have substantially extended the discussion to argue that computational requirements (especially with regard to memory footprint) are very different for forward versus adjoint models, and that for this reason (mainly, but not exclusively) the preferred option is to disable code at differentiation time. The argument is repeated here, and goes as follows:**

*Like many complex, time-evolving geophysical models, SICOPOLIS comes with a range of choices of model configuration, in particular numerical schemes, which the user may choose from. As a matter of convenience, the preferred implementation is to make all of these choices (or options) available at runtime, such as to minimize the need for recompiling the model. The same convenience is25available, in principle, to the AD-generated adjoint model. The control flow analysis of the AD tool identifies all possible flows of forward model execution and produces corresponding adjoint flow paths. However, close to two decades of experience with the application of AD to complex, time-evolving geophysical models, all of which have a range of numerical schemes that users may choose from (Heimbach et al., 2002, 2005; Forget et al., 2015), has shown that for the specific application of adjoint modeling, it is preferable to remove code that will not be executed in a given application from adjoint code generation (and subsequent compilation). The two main reasons for proceeding in this manner are:*

(i) *Exclusion of forward model code that the user knows will not be executed may significantly simplify the AD tool's dependency and flow control analysis, avoid spurious dependencies that the AD tool may detect, and lead to more streamlined source code for the adjoint;*

(ii) *Because of the reverse mode and requirement to store required variables in time-reversed order (e.g., those used for evaluating state-dependent conditions and nonlinear expressions), adjoint models will have a substantially larger memory footprint than their parent forward model (Heimbach et al., 2005). Memory requirements may be significantly increased if the adjoint model is required to keep track of a large range of conditional branches for execution.*

*For these practical considerations, removing non-used forward model code at the time of adjoint code generation and subsequent compilation has proven to be highly preferable (although not strictly required). It is implemented here via C preprocessor (CPP) options*

*that are enabled or disabled prior to generating the adjoint code (and prior to compilation time. We note that the implementation keeps runtime parameters and flags in place, such that the forward model default to keep all code available at runtime is not compromised. By pairing SICOPOLIS with source-transformation tool OpenAD, the adjoint model of SICOPO-10LIS may be generated automatically, for a large variety of forward model configurations (including detailed choices of model domain, numerics, as well as control variables and QoI).*

P10, L3 : Does this raise the question about why, in the adjoint, some time steps are more stable than others? In other words, why can't one take the same time step sizes than in the forward simulation. That can be an interesting question for a Numerical Analysis specialist (not me...)

**This is a well-spotted error on our part: we take that time step because it ensures stability in the forward model, *not* the adjoint model. We re-wrote for clarity.**

P10, L31: I'd replace "sufficently approximates" with "is sufficiently consistent with", because I tend to think that it is divided differences that is an approximation of the other.

**Done.**

P11, Table 1: I would swap rows 7 and 8 for consistency with rows 3 and 4.

**Done.**

*Deviations on rows 5 to 8 seem surprisingly high. Are they discussed in the text ? I read in P14 L20 that the finite difference chosen is around 5%, which can explain the high deviation. And yes, the explanation in P15 L1 may be right. But does the deviation decrease when the divided difference is smaller e.g. 0.5% instead of 5% ? As it is, a deviation of 57% is still worrying.*

**We discuss in somewhat more detail the mismatch of these finite volume and adjoint calculations on P15 L14. In general, the temperature control variables display a greater mismatch between adjoint and finite difference than the other control variables. The two main issues are (1) numerical noise when sensitivities are very small compared to the QoI, which affects the finite difference; and (2) n SICOPOLIS, the thermal equations employ a greater number of non-differentiable terms. We note this in the revised version, and point also to the appendix for a discussion on non-differentiable code. As for the choice of 5% deviation: this value was not only selected in accordance with a previous adjoint model of SICOPOLIS (Heimbach and Bugnon, 2009), but also because deviations < 5% did not result in appreciable changes to the cost function at all.**

*P15 L3: The question that comes immediately is how do these times compare with the primal simulation ? It might also be appropriate to describe the checkpointing scheme used in this*

*experiment, on time-stepping: is it multi-level, or binomial, how many checkpoints are used, how many duplicated forward steps. Are these questions left for future work? Oh, I see it is in the appendix, P24 L24. Could you just, in the main text, point out that the appendix mentions that ?*

**Indeed, it's described in Appendix B. Reference to it is now added.**

*P17 L17: The question of best practices makes me think that you may want to cite the Utke-Hascoet paper on that "Programming language features, usage patterns,..."*

**Done (thanks!)**.

(OMS 2016)
Technical corrections:
**Done:** P2, L25 : "construed"
**Done:** P4, L13 : "more confident projections" -> "more faithful" ?
**Done:** P5, L30 : "substantive ... to" -> "substantial ... over" ?
**Done:** P6, L1: "ever" or "even" ?
**Done (deleted):** P8, L15: why comma after warmer?
**Done:** P9, L25: "appendix" is repeated
**Done:** P10, L13: "instantaneously" ? "instantly"
**Done (well-spotted):** P13, L3: Is it January? Figure 3 caption writes July. Or did I miss something?
**Done:** P16, L18: "introduces" or "-introduced" ?

---

## Author Comment (AC2) · 1 Dec 2019

**We would like to thank the reviewer for her careful read of the manuscript and constructive comments. In the following, we respond comment-by-comment (our replies are in bold-face).**

General comments: The manuscript by Logan et al. describes the generation of an adjoint code by algorithmic differentiation (AD) for the ice sheet model SICOPOLIS. The authors give a clear explanation of the motivation for adjoint modelling and for the use of AD in generating adjoint code, and they produce and interpret simple example applications for both Antarctica and Greenland.

I thank the authors for producing a well written and easy to follow manuscript. As a C1 non-expert in adjoint modelling, I found P3 an excellent description of its context and capabilities. I found the demonstration sensitivity analysis in Figures 2-3 interesting as well.

I am not the best person to comment on validation of the adjoint vs. finite-difference code, but I found the authors' explanations generally supportive of their conclusions. Perhaps a bit more data (e.g. absolute value of the QoI) could be given to support the comment about numerical noise producing high misfit in Table 1, Columns 5 and 6.

I agree with the authors that understanding uncertain input variables and models' sensitivity to them is important for contextualizing ice sheet/sea level projections. However, I disagree with the framing of the first paragraph of the introduction, namely that (A) effective adaptation to/mitigation of sea level rise relies on reducing uncertainty in projections and (B) development of more sophisticated ice sheet models will help reduce uncertainty.

Regarding (A): The social-science literature of climate adaptation discusses assorted factors that affect adaptive capacity, many of which have little to do with the state of the science. If the authors are interested, they could refer to e.g. Lemos and Rood 2010 (WIREs Climate Change) for a discussion of the "uncertainty fallacy" in climate science.

Regarding (B): It is intuitive that improved understanding of ice sheet dynamics will help us produce models that give more physically-consistent ("predictable") results. But physical consistency does not always translate to less uncertainty. For example, models that include "tipping point" dynamics (or hysteresis and multiple steady states) are arguably more sophisticated than those that do not, yet future projections over a range of climate scenarios may show a wider, not narrower, range when tipping point dynamics are included. Initial efforts to improve model sophistication by including newly-understood or newly-proposed processes can also increase uncertainty in terms of inter-model or inter-scenario spread. A notable example is the widening of 21st-22nd C2 century sea level projections shown in DeConto Pollard 2016 when the dynamics of marine ice cliff collapse and hydrofracture were included.

*In a revised version, I suggest the authors strengthen the framing of the first paragraph to focus on the need for context and improved understanding, rather than leaning on the uncertainty angle.*

**These are really interesting points and We are glad you brought them up.**
**Regarding (A): Ok. Have removed the first sentence.**

**Regarding (B): you are correct, this work does not necessarily contribute to *reducing* uncertainties, but instead to provide a more complete characterization of uncertainties through calculation of comprehensive model sensitivities (which goes toward quantifying parametric and initial condition uncertainties). We have clarified this point in the introduction, and now omit the previously stated goal of "reducing uncertainties".**

**We note that the purpose of the introduction is to set the stage for the following discussion on the need to understand how \*a particular\* model responds to perturbations. Equipped with an adjoint model, their behaviors can be interrogated more rigorously, leading to the improved context and understanding needed to appreciate their projections. We also note that this work does not actually perform any predictions (or projections).**

The revisions I suggest to the introduction and in the specific comments below are relatively minor and should not impede publication. I imagine that many ice-sheet modellers will be interested in what the authors have shown here, and I look forward to reading follow-up studies using SICOPOLIS-AD.

Specific comments:

Figures 2 and 3 - both figure captions state that the [B] subplots illustrate sensitivities to July temperature. The text on P13,L3, Table 1, and later discussion refers to January or "summer" precipitation. Is there a mistake in the figure captions, or do the figures depict something not discussed in the text?

**Good catch. The captions said "July" in error. We corrected it to "January". All the tests were for summer (January for Antarctica, July for Greenland).**

P4,L11 - "A model that can..." i.e. a forward model that can achieve the state deemed optimal by the Lagrange multiplier method? Does "reproduce the optimal behavior" refer to a model-vs-model or a model-vs-observation comparison?

**Clarified: we meant model-vs-observation.**

P5,L10 - Is the adjoint code acceptable if the finite-difference-derived sensitivities approximate the adjoint-derived sensitivities, or is it the other way round? Intuitively I would expect that we

accept the adjoint code if the sensitivities it produces approximate those derived by finite difference; that is, the finite-difference sensitivities are the "standard" against which the adjoint code is judged.

**In theory, it is the other way round. AD applied to discretized code produces the derivative of this implementation to very high precision, unlike finite-differencing, which depends on the order of the finite-differencing scheme and the epsilon chosen (the monograph by Griewank and Walther, 2008, which we cite, discuss this point in great detail). In practice, and for the purpose of this work, we wish to test and ascertain, that the AD tool has produced "correct" adjoint code, for which we use finite-differencing as a reference, but acknowledge that accuracy may be lower.**

P5, Eq 3 - Given the explicit mention of "tolerance" in line 10, I might write the righthand side of this equation with $= \partial J \, \partial x + \delta$, where $\delta$ is the accepted tolerance.

**There are two issues with writing it this way: (1) the tolerance so defined is unit-dependent (and thus may change for different physical variables of the extended control vector); (2) for very small sensitivities, the F.D. may pick up numerical noise, leading to large relative differences to the adjoint-generated derivative. Since our goal here is not to implement high-order F.D. schemes, we prefer to leave the discussion at this elementary level. Ultimately, judicious (or cautious) application of adjoint sensitivities in detailed studies should re-affirm the adjoint to be accurate (in the oceanographic context, please see e.g., Pillar et al. 2016; Smith and Heimbach 2019).**

P5,L18 - This is a dense list of references without much discussion. Given that this manuscript focuses on an ice-sheet application, the study might be well-served by adding another paragraph to discuss specific distinctions among these past efforts in adjoint modelling of ice sheets and any notable contrasts with the present study.

**Following your suggestion, we separated the reference, based on different applications (we still kept it brief, as the purpose of this paper is not to provide a review of the subject).**

P10,L2-5 - What is the initial geometry? It is a bit unclear whether the experiment is 100 yr near equilibrium, a 100-yr spin-up, or something else. I don't know that it matters for the adjoint process, but it would be nice to have some more clarity.

**Good questions. The revised version clarified this. We use the Antarctic ice sheet geometry of Fretwell et al. (2013), now clarified near the end of the paragraph.**

P13, Fig 3 - I understand that the point of showing the logarithm of the absolute value of the sensitivities is so that the reader can compare their order of magnitude, both across parameters and within-parameter spatial variability. Is there a reason that each subplot uses a slightly different colormap? Could one colormap be applied to all subplots to facilitate intercomparison?

**The subplot intercomparison is indeed made more difficult when the color bounds are not uniform across the panels. Unfortunately, given the large value differences between the different control runs, pattern would simply be lost. As for the blue-red vs blue-yellow color maps, these were selected with purpose: blue-red is a natural fit for positive-negative, where the absence of color (white) matches with zero. Because the other plots are meant to display overall pattern, we wished to reserve the white, zero-centered scale for the positive-negative patterns.**

P17,L4-6 - Is there a use for adjoint modelling in distinguishing between different possible parameterizations of these processes?

**Yes! Absolutely a use for adjoint models there, and we include this now.**

Technical comments:

**Done:** P2,L23 - For balance, an example of a quantity that "parameterize[s] subgrid-scale processes or empirical constitutive laws" would be helpful. Perhaps iceberg calving or the routing of surface/basal meltwater would be an appropriate example to include?

**Done:** P2,L30 - "key quantities of interest that represent integrated quantities of an ice sheet" -> "ice-sheet-integrated quantities of interest"?

**Done:** P9,L22 - "...inherently non-differentiable, sometimes required..." is a comma splice. Replace comma by "and"?

**Done:** P16,L11 - "Thus ... within the main trunk of SICOPOLIS" is a very long sentence and I had to read it multiple times to understand. Consider streamlining.

**Done: [re-phrased]** P16,L18 - The phrase in the parentheses is hard to parse and might be a run-on. The authors might consider replacing the remarks in parentheses with another full sentence or two to flesh out the thought.

**Done: [good observation]** P20,L1 - "Precipitation" or the "sensitivity to precipitation" is almost entirely positive?

**Done:** P20, L3 - Typo "qsimulations"

References cited:

--

Pillar, H. R., Heimbach, P., Johnson, H. L., & Marshall, D. P. (2016). Dynamical Attribution of Recent Variability in Atlantic Overturning. Journal of Climate, 29(9), 3339–3352. http://doi.org/10.1175/JCLI-D-15-0727.1

Smith, T., & Heimbach, P. (2019). Atmospheric Origins of Variability in the South Atlantic Meridional Overturning Circulation. Journal of Climate, 32(5), 1483–1500. http://doi.org/10.1175/JCLI-D-18-0311.1

---

## Referee Report (RR1)

The paper is for the most part well written and I recommend it for publication after minor revisions. My major concern is the big discrepancy (up to 50% relative error) between the adjoint-based and finite-difference based sensitivities shown in Table 1. This seems large even considering numerical errors or high nonlinearities. The finite-difference based sensitivity depends heavily on the increment $\varepsilon$ used, especially for highly nonlinear forward maps. Maybe the authors could consider trying different increments.

Detailed review:

Eq (3) In general $x$ is not a scalar, so the definition provided here is not well defined. I would change it to something like $\delta_\varepsilon \mathcal{J} = \frac{\mathcal{J}(x+\varepsilon\delta x) - \mathcal{J}(x-\varepsilon\delta x)}{2\varepsilon}$ This is also the proper finite difference approximation of $\delta\mathcal{J}$ introduced in eq (6).

Eq (4) It is not clear why $\mathcal{L}_0$ has argument $x$ (the control) while $\mathcal{L}_1, \mathcal{L}_2, \ldots$ seem to have the state $u$ as argument. Is it assumed that the initial state $u_0$ is the control $x$? If so please mention this. This would be a simplified case because in the following you use the precipitation, surface and basal temperature (which are not an initial states) as a control.

Page 9, Line 14 Note that also the MALI model (Hoffmann et al.) and the FEIS model (Brinkerhoff et al.) use AD, through Trilinos and FEniCS softwares respectively.

Fig 1 I think this figure is hard to understand. What is $dy$? Why is it set to zero at some point in the reverse sweep? Can you please make the figure and its caption clearer?

Page 11, Line 14 This calving law can create issues with computing sensitivities, because the thickness becomes discontinuous in time.

Table 1 I think that the notation $\frac{\delta\mathcal{J}}{\delta\text{variable}}$ is misleading. In my understanding here you are showing $\delta\mathcal{J}$, computed as in (6), where $x$ is the variable and $\delta X = \varepsilon e_i$, $e_i$ being the unit vector that's equal to 1 at point $i$ (for the cases 1,2,3) and $\varepsilon$ is 5% of the initial variable value. Similarly the notation $\frac{\Delta J}{\Delta\text{variable}}$ is misleading.

Page 15, Line 21 The definition of $\Delta V_{adj}$ is a bit problematic. Instead of $\frac{\delta\mathcal{J}}{\delta X}$ it should be $\frac{\partial\mathcal{J}}{\partial x}$. Moreover the integral makes sense only in the continuous case. I think it is better to use the notation of eq. (6) and write $\Delta V_{adj} = < \frac{\partial J}{\partial X}, \delta X >$, where $\delta X = \varepsilon\chi_\Omega$ and $\chi_\Omega$ is the characteristic function of $\Omega$, i.e. it's 1 on $\Omega$ and 0 otherwise, and $\varepsilon$ is 5% of the variable value. In the discrete case $< \cdot, \cdot >$ is simply the $l_2$ inner product, i.e. a sum over all the nodes of the grid. Does this correspond to how you computed $\Delta V_{adj}$?

Page 15, Line 22 In the definition of $\Delta V_{fd}$, at the denominator there should be only 2, not $2\delta X$.

Appendix  While automatic differentiation of functions having corners (e.g. the absolute value or max function) makes sense, the differentiation of discontinuous functions does not makes sense mathematically (one would have to deal with the Dirac delta). For this reason I'm concerned to see the AD implementation of discontinuous functions like floor or ceiling. Why are those function needed in an ice sheet code?

---

## Author Response (AR2)

Dear Alex,

we are happy to submit a revised manuscript that takes into account all comments/suggestions from the latest round of reviews. Also provided are responses to reviewers and a latexdiff-ed version of the manuscript.

We have also no created a non-revocable repository of the SICOPOLIS-AD v1 code on Zenodo (https://doi.org/10.5281/zenodo.3686393) to comply with GMD guidelines, as documented in the manuscript.

Thank you & best regards,
Patrick

Response to reviewer Liz Ultee:

We thank the reviewer again for her careful examination of the manuscript and her comments. Answers to her comments are provided in the following.

> *The authors have, at my suggestion, added text on manuscript P5 to separate previous applications of adjoint sensitivities in ice sheet modelling. I agree with the authors that the purpose of this manuscript is not to provide a review of the topic. However, my original suggestion was to discuss the previous work in more detail so that \*any notable contrasts with the present study\* (i.e. the novelty of this manuscript) might be drawn out. In my opinion, the new text on P5, L22-33 does better than the original to point interested readers to related work, but it does not quite achieve the aim of highlighting the scientific contribution of this manuscript.*
>
> *I apologize if my original suggestion was unclear. If the work will be revised again (which I leave to the discretion of the editor/authors), the authors might consider just a little more wordsmithing on this paragraph to better integrate the discussion of previous work with signposting toward the manuscript's novelty. In this present version, the authors do add some phrasing later in the manuscript that helps highlight novelty and applications. This is acceptable.*

As the reviewer notes, we have addressed novel aspects of the work later in the manuscript. Nevertheless, we appreciate the suggestion to better integrate this aspect within the discussion of existing work. We have followed her suggestion to modify the paragraph.

> *Technical corrections*

We thank the reviewer for her suggestions and pointing out errors. We have implemented them all in the revised manuscript.

Response to reviewer Mauro Perego:

We thank the reviewer for the careful examination of the manuscript and his comments which helped to improve the manuscript. Answers to his comments are provided in the following.

> *General comment: "My major concern is the big discrepancy (up to 50% relative error) between the adjoint-based and finite-difference based sensitivities shown in Table 1. This seems large even considering numerical errors or high nonlinearities. The finite-difference based sensitivity depends heavily on the increment " used, especially for highly nonlinear forward maps. Maybe the authors could consider trying different increments.*

We agree with the reviewer that these differences appear large. They are however, not related to step sizes in the finite difference approximation (we have investigated this dependency). We had elaborated on the possible causes of these differences in response to a previous reviewer, and repeat it here for the sake of completeness. The issue merits further investigation as the model is applied to more comprehensive studies:

"We discuss in somewhat more detail the mismatch of these finite volume and adjoint calculations on P15 L14. In general, the temperature control variables display a greater mismatch between adjoint and finite difference than the other control variables. The two main issues are (1) numerical noise when sensitivities are very small compared to the QoI, which affects the finite difference; and (2) n SICOPOLIS, the thermal equations employ a greater number of non-differentiable terms. We note this in the revised version, and point also to the appendix for a discussion on non-differentiable code. As for the choice of 5% deviation: this value was not only selected in accordance with a previous adjoint model of SICOPOLIS (Heimbach and Bugnion, 2009), but also because deviations < 5% did not result in appreciable changes to the cost function at all."

> *Eq. (3): In general, x is not a scalar, …*

We thank the reviewer for pointing this out. We realize that in an attempt to simplify equations as much as possible, accuracy of what is expressed was compromised. We have corrected this.

> *Eq. (4): It is not clear…*

Again, we simplified too much here, compromising accuracy. We have clarified this, modified equation and accompanying text accordingly.

> *Page 9, Line 14: MALI and VarGlas adjoints*

Thank you for pointing these out, we have added these references as examples of operator-overloading and code-templating AD.

> *Fig. 1: Increase clarity.*

We have improved both the figure caption as well as the main text to explain what is shown

> *Page 11, Line 14: The calving law…*

The calving law represents an example of non-smooth code behavior described in Appendix B. To address the reviewer's comments, we have added a short paragraph on how AD treats this case:

"We note that the operation underlying calving (see above) amounts to a conditional statement. From an AD perspective, the following steps occur: (i) derivative codes generated for each condition; (ii) code to store and restore of required variable is added to properly evaluate the conditional derivative. For legacy code the operation may not be differentiable at the exact condition (see Appendix B for practical details). This should be taken into account when performing gradient-based optimization."

*Table 1, Notation*

We have modified the notation along with the Table caption to provide more clarity what is presented. (We note that the top row in this table is meant as an abbreviation only in order to fit within the table formatting).

*Page 15, Line 21: Definition of V_adj …*

We have modified this definition, but think that the presentation as an integral is sufficiently clear within this context.

*Page 15, Line 22: V_fd*

Corrected, thank you for spotting.

*Appendix: AD for non-smooth code.*

Much can be written about non-smooth functions, in particular those implemented in legacy code. We attempted to describe circumstances where such functions may be encountered, some of which are benign (index evaluations) some of them depending on how they are evaluated and subsequently used in an actual implementation (e.g., well-defined one-sided finite-differences). We have also described how to alleviate the issue for legacy code. There is an emerging literature in the machine-learning community, where "backpropagation" (akin to adjoints) are used and requiring differentiable programming. The reality of non-smooth functions and issues therewith are not limited to the use of AD. AD merely exposes such issues.

[revised manuscript text omitted]